# The Effect of N-Acetylation on the Anti-Inflammatory Activity of Chitooligosaccharides and Its Potential for Relieving Endotoxemia

**DOI:** 10.3390/ijms23158205

**Published:** 2022-07-26

**Authors:** Wentong Hao, Kecheng Li, Xiangyun Ge, Haoyue Yang, Chaojie Xu, Song Liu, Huahua Yu, Pengcheng Li, Ronge Xing

**Affiliations:** 1CAS and Shandong Province Key Laboratory of Experimental Marine Biology, Center for Ocean Mega-Science, Institute of Oceanology, Chinese Academy of Sciences, Qingdao 266071, China; haowentong0723@163.com (W.H.); 17346872166@189.cn (X.G.); yanghaoyue@qdio.ac.cn (H.Y.); chaojiexu725@163.com (C.X.); sliu@qdio.ac.cn (S.L.); yuhuahua@qdio.ac.cn (H.Y.); pcli@qdio.ac.cn (P.L.); 2Laboratory for Marine Drugs and Bioproducts, Pilot National Laboratory for Marine Science and Technology (Qingdao), No. 1 Wenhai Road, Qingdao 266237, China; 3University of Chinese Academy of Sciences, Beijing 100049, China; 4School of Marine Science and Engineering, Qingdao Agricultural University, Qingdao 266109, China

**Keywords:** chitooligosaccharides, degree of acetylation, anti-inflammation, endotoxemia

## Abstract

Endotoxemia is mainly caused by a massive burst of inflammatory cytokines as a result of lipopolysaccharide (LPS) invasion. Chitooligosaccharides (COS) is expected to be a potential drug for relieving endotoxemia due to its anti-inflammatory properties. However, the structural parameters of COS are often ambiguous, and the effect of degree of acetylation (DA) of COS on its anti-inflammatory remains unknown. In this study, four COSs with different DAs (0%, 12%, 50% and 85%) and the same oligomers distribution were successfully obtained. Their structures were confirmed by ^1^H NMR and MS analysis. Then, the effect of DA on the anti-inflammatory activity and relieving endotoxemia potential of COS was researched. The results revealed that COS with a DA of 12% had better anti-inflammatory activity than COSs with other DAs, mainly in inhibiting LPS-induced inflammatory cytokines burst, down-regulating its mRNA expression and reducing phosphorylation of IκBα. Furthermore, this COS showed an obviously protective effect on endotoxemia mice, such as inhibiting the increase in inflammatory cytokines and transaminases, alleviating the injury of liver and intestinal tissue. This study explored the effect of DA on the anti-inflammatory activity of COS for the first time and lays the foundation for the development of COS as an anti-inflammatory drug against endotoxemia.

## 1. Introduction

Endotoxin is a hydrophobic domain of bacterial lipopolysaccharide (LPS). LPS is a main constituent of gram-negative bacterial membrane, the crucial pathogen-associated molecular patterns to activate toll-like receptor-4 (TLR4) and it can regulate inflammatory and innate and adaptive immune responses [1,2,3]. When large amounts of endotoxin released into the blood, it can induce endotoxemia, also known as sepsis. Endotoxemia can cause damage to multiple organs such as the liver [4], the kidney [5], the intestines [3] and the heart [6] etc. For example, endotoxins can trigger intestinal inflammatory diseases, causing increased hepatic inflammatory cell infiltration and renal tubular cell damage, etc. At the same time, when a large amount of endotoxin accumulates in the body and exceeds the balance that the body can control, it will cause massive burst of pro-inflammatory cytokines such as interleukin (IL)-6, tumor necrosis factor (TNF)-α, IL-1β, and lead to endotoxic shock accompanied by disseminated intravascular coagulation [7,8,9]. More seriously, not only are the multiple organs of patients with endotoxemia caused by bacterial infections more susceptible to damage, but patients with other diseases also have a higher rate of reinfection with endotoxemia [10]. In recent years, there have been more and more patients with endotoxemia-related acute kidney injury, acute liver injury and other endotoxemia-related diseases, which have seriously affected human health. Commonly used anti-inflammatory drugs are divided into steroid anti-inflammatory drugs and non-steroidal anti-inflammatory drugs. Long-term or large-scale use of steroid anti-inflammatory drugs such as glucocorticoids often causes metabolic disturbances and impairs normal programmed responses. Non-steroidal anti-inflammatory drugs also tend to cause adverse effects such as gastrointestinal complications, renal insufficiency, and cardiovascular disease [11,12]. Therefore, it is necessary to find a non-toxic, green drug as soon as possible to better solve this type of acute inflammation. 

Chitooligosaccharides (COS) is the product obtained by chemical or enzymatic degradation of chitosan that is commonly extracted from shells of marine crustaceans. COS has smaller molecular weight (MW), lower degree of polymerization (DP), and good water solubility which chitosan does not possess [13,14,15]. Similar to other natural carbohydrate [16,17], COS has been shown to have good immunogenicity and anti-inflammatory activity in vitro [18,19,20] and in vivo [21,22,23], and is expected to be a good natural anti-inflammatory drug to prevent and treat endotoxemia. The anti-inflammatory activity of COS is related to its structural parameters, including MW, DP and degree of acetylation (DA) etc. It is important to study the influence of these structural parameters on its anti-inflammatory activity for the early application of COS as an anti-inflammatory drug. In carrageenan-induced paw edema model, COS can play an anti-inflammatory role by inhibiting the synthesis of cyclooxygenase and reducing the secretion of prostaglandins, and COS with MW of 1.2 kDa produces better anti-edematous effect than that of 5.3 kDa. The efficacy of high-dose COS with MW of 1.2 kDa was comparable to that of indomethacin [24]. Zhao et al. further reported that COS with DP > 4 had better anti-inflammatory activity and chitoheptaose was found to be the optimal component with antioxidant, anti-inflammatory and anti-apoptotic activities in a model of myocarditis [6]. Additionally, the DA of COS also have important effect on its bioactivity. P. Santos-Moriano et al. prepared three COSs with MWs between 0.2 kDa and 1.2 kDa with different DAs and compared their ability to reduce the production of TNF-α [25]. However, when investigating the influence of COS with a certain property on its bioactivity, other parameters of COS should remain the same. Previous reports on the anti-inflammatory activity of COS, such as P. Santos-Moriano’s work, rarely control both DA and DP factors simultaneously. Therefore, the DA effect of COS on its anti-inflammatory activity is still unclear.

In this study, a series of COSs with different DAs and the same DP distribution were obtained by N-acetylation reaction and graded alcohol precipitation. The component structures of these COSs were confirmed by nuclear magnetic resonance (NMR) and mass spectrometry (MS). Then, the mouse macrophages RAW264.7 were used to evaluate the effect of DA on the anti-inflammatory activity of COS, and the optimal COS with well-defined DA was selected to evaluate the protective effect on endotoxemia mice.

## 2. Results

### 2.1. Preparation and Characterization of COSs with Different DAs

The preparation of a series of COS mixtures ranging from DP 2 to 6 with DAs varying from 0% to 85% was carried out by N-acetylation reaction and graded alcohol precipitation. Their structures were characterized with ^1^H NMR. As is shown in Figure 1, the ^1^H NMR analysis confirmed that all repeating units of the COS from graded alcohol precipitation was fully N-deacetylated, because no signal at 1.9–2 ppm corresponding to methyl protons of the N-acetyl groups was present in the spectrum. Therefore, this product is COS with DA = 0%. Other COSs of different DAs were preparate by the N-acetylation of the original COS. The signal intensity at 1.9 ppm increases after N-acetylation, while the signals of glucosamine (GlcN) H-2 around 2.8–3.2 ppm decrease. The determination of the average DA of the different partially N-acetylated COSs was performed using signal areas of H-2 protons of GlcN units (A_GlcN H-2_) and acetyl protons of N-acetylglucosamine (GlcNAc) units (A_CH3_) according to the method in previous study [26]. The DAs of four prepared COS samples are 0%, 12%, 50% and 85%, respectively. (In the following, we use 0% COS, 12% COS, 50% COS and 85% COS to represent these four COSs, respectively.)
(1)$$\mathrm{DA}\left(\%\right)=\frac{\frac{1}{3}A_{\mathrm{CH}3}}{\frac{1}{3}A_{\mathrm{CH}3}{+A}_{\mathrm{GlcN}\;H-2}}\times 100$$

High performance liquid chromatography (HPLC) analysis proved these four COSs has the same DP distribution, both ranging from 2 to 6. (As shown in Appendix A). Due to the lack of standards for partially acetylated monomers, the component analysis of COS samples was further determined by electrospray ionization mass spectrometry (ESI-MS). The results are shown in Figure 2, A–D corresponding to COSs with DA of 0%, 12%, 50%, and 85%, respectively. We use **A** to represent GlcNAc unit, **D** to represent GlcN unit and Arabic numerals after the monosaccharide unit to indicate the number of monosaccharides. As shown in Figure 2A, the main components of 0% COS are (GlcN)_2–6_. The ion peaks at *m*/*z* 171.08 and 341.16 correspond to (GlcN)_2_; The ion peaks at *m*/*z* 251.62 and 502.22 correspond to (GlcN)_3_; The ion peaks at *m*/*z* 221.77, 332.15 and 663.29 correspond to (GlcN)_4_; The ion peaks at *m*/*z* 275.46 and 412.68 correspond to (GlcN)_5_; The ion peak at *m*/*z* 493.22 correspond to (GlcN)_6_. Moreover, there is no ion peak corresponding COS with acetyl groups in it. The 12% COS has similar DP distribution with 0% COS, ranging from 2 to 6. Furthermore, the 12% COS also contains some other N-acetylated oligosaccharides, such as trisaccharide with one acetyl group (**D2A**, *m*/*z* 544.24), tetrasaccharide with one acetyl group (**D3A**, *m*/*z* 705.31), pentasaccharides with one acetyl group (**D4A**, *m*/*z* 433.69), and a small amount of trisaccharides with two acetyl groups (**DA2**, *m*/*z* 586.25). As shown in Figure 2C, when the COS was further acetylated and the DA reached 50%, the monosaccharides with two or even three acetyl groups increased. The main components of 50% COS are **D2A** (*m*/*z* 272.62, *m*/*z* 544.23), **D2A2** (*m*/*z* 374.16, *m*/*z* 747.31), **D3A2** (*m*/*z* 454.69), **D3A3** (*m*/*z* 556.23), while the oligosaccharides without acetyl group nearly do not exist. Finally, when the DA reaches 85%, it mainly contains some highly acetylated oligomers including **DA2** (*m*/*z* 586.24), **DA3** (*m*/*z* 789.32), **D2A3** (*m*/*z* 475.70), **DA4** (*m*/*z* 992.40) (as shown in Figure 2D), and some fully acetylated products (chitin oligosaccharides) also appear, such as the ion peak at *m*/*z* 650.24 corresponds to (GlcNAc)_3_; the ion peak at *m*/*z* 853.32 corresponds to (GlcNAc)_4_. The mass spectrometry results prove that the DP distribution of our COS series samples are consistent, and these COSs can be used to evaluate the effect of acetylation on anti-inflammatory activity.

### 2.2. Effects of COSs with Different DAs on Inflammatory Cytokines

Now that a series of COS samples with different DAs have been prepared, the effect of DA on their anti-inflammatory activity was further investigated. Firstly, the cytotoxicity experiments of these COSs were performed. RAW 264.7 cells were incubated with COSs for 48 h, followed by 3-(4,5-dimethyl-2-thiazolyl)-2,5-diphenyl-2-H-tetrazolium bromide (MTT) assay. There was no significant difference in cell viability during COSs incubation between each group and the negative control (NC) group(as shown in Figure 3), indicating that these COSs had no cytotoxic effect on cells at the concentrations used in this study and the treatment of COSs did not change the cell morphology. In addition, high concentrations (800 μg/mL) of COS could cause cell proliferation. Then, an appropriate concentration was selected to explore the effect of COSs with different DAs on LPS-induced inflammatory response. As shown in Appendix A, LPS significantly increased the secretion of inflammatory cytokines and the expression of their mRNA levels. The cells treated with COS can effectively reduce the adverse effects of LPS in a concentration-dependent manner. When the concentration reaches 600–800 μg/mL, it can even directly eliminate the effect of LPS. Since high concentrations (800 μg/mL) of COS have cell proliferation effects, the concentration of 600 μg/mL was chosen in the following experiments. Overall, the concentration of COSs used in this study had neither cytotoxic nor cell proliferation-promoting effect on cells.

Macrophages play an important role in the innate immunity and acquired immunity. When macrophages are activated by LPS, they produce NO and various cytokines, such as TNF-α, IL-6 etc. [27]. Therefore, we first measured the effects of COSs with different DAs on LPS-induced bursts of these inflammatory cytokines through enzyme-linked immunosorbent assay (ELISA) kits. As shown in Figure 4, LPS significantly increase the secretion of these inflammatory cytokines. All COS samples could reduce the sharp increase in NO content caused by LPS and 12% COS showed the best effect among them, which has a significant difference compared with other COSs. There was no significant difference between the NC group (which only added RPMI medium) and the 12%group (12% COS + LPS), which means 12% COS can even directly eliminate the NO changes caused by LPS. LPS can also significantly increase the inflammatory cytokines IL-6 and TNF-α. For TNF-α, 0% COS, 12% COS, and 85% COS can reduce the influence of LPS, and 12% COS show the best activity. However, as for IL-6, only 12% COS can eliminate the effect of LPS and make the content of IL-6 back to normal levels.

Based on the ability of COS to inhibit LPS-induced inflammatory cytokines burst, these related gene expressions (iNOS, IL-6 and TNF-α) were assessed correspondingly by quantifying the mRNA expression levels of these cytokines. As shown in Figure 5, 12% COS can significantly down-regulate the expression of inducible nitric oxide synthase (iNOS), IL-6 and TNF-α genes caused by LPS, followed by 50% COS. 0% COS and 85% COS can only down-regulate the gene expression of iNOS and IL-6. At the same time, the expression of several other genes closely related to inflammation were also detected, such as tumor-related cyclooxygenase-2 (COX-2) [28], oxidative stress-related nicotinamide adenine dinucleotide phosphate (NADPH) oxidase (NOX2) [29] and inflammasome-related IL-1β [30,31]. Quantitative analysis of these genes is also shown in Figure 5. The results showed that the effects of COS on these genes expression are also dependent on its DA. All COS samples can reduce the upregulation of NOX2 gene caused by LPS, and 12% COS inhibits the effect more significantly. However, for the up-regulation of COX-2 and IL-1β gene expression caused by LPS, the fully deacetylated COS and the COS with low DA (DA = 12%) showed better inhibitive effect. The highly acetylated COS (DA = 85%) even display a promotion effect on the secretion of IL-1β, suggesting a little pro-inflammatory activity.

### 2.3. Effects of COSs with Different DAs on NF-κB Signaling Pathway

Since COS can alleviate the inflammatory symptoms caused by LPS in inhibiting inflammatory cytokines way. Moreover, as a classic signaling pathway of inflammation, NF-κB regulates the secretion of inflammatory cytokines [32,33]. The differential effects of COSs with different DAs were then explored on protein expression and modification in the NF-κB signaling pathway. NF-κB is often associated with the anti-inflammatory mechanism of carbohydrates [19,34]. In the resting state, NF-κB is located in the cytoplasm and forms a complex with the inhibitory protein IκBα. LPS can activate IκB kinase (IKK). IKK in turn phosphorylates IκBα, which leads to ubiquitination of the IκBα, allowing IκBα to be detached from NF-κB, and ultimately IκBα is degraded by the proteasome. Activated NF-κB (p65) then translocate into the nucleus, causing an increase in many inflammatory cytokines [35]. As shown in Figure 6, after the cells were lysed, the total protein was extracted for western blot (WB) experiments. Results showed that the phosphorylated-IκBα/IκBα ratio in LPS group was significantly higher than that in the NC group, indicating NF-κB pathway activation and inflammation burst. In contrast, 12% COS and 50% COS can reduce the phosphorylation of IκBα induced by LPS, which means that DA is involved in inhibiting the activation of NF-κB signaling pathway and 12% COS showed the best effect on inhibiting the phosphorylation of IκBα. Similar to the IL-1β gene expression level results mentioned above, when the DA was 85%, the highly N-acetylated COS showed a promotion effect of the IκBα phosphorylation, suggesting some pro-inflammatory activity to a certain extent similar to LPS.

Then, the immunofluorescent staining assay was used to analyze the nuclear translocation of NF-κB in 12% COS group, as shown in Figure 7 (This result is for qualitative analysis only). When p65, which fluoresces green, enters the blue nucleus, the two colors merge to produce a bright cyan color. Additionally, the cyan color can be seen in the LPS group, which is prominent and bright. While the green in the NC group is mostly concentrated outside the nucleus, and the 12% COS treatment group also has cyan, but the cyan is not as obvious as the LPS group and the green fluorescence around the nucleus can still be seen. It shows that 12% COS could effectively inhibit the activation of NF-κB signaling pathway by preventing p65 entry into the nucleus to a certain extent. Taking all these above assays into account, DA is proved to be an important structural parameter influencing the anti-inflammatory activity of COS and the 12% COS is the optimal one among all four COS samples with different DAs. Therefore, we selected the COS with DA of 12% to investigate the protective effect on the endotoxemia mice and evaluate the potential as an anti-inflammatory drug.

### 2.4. Alleviating Effect of 12% COS on Endotoxemia in Mice

Endotoxemia causes inflammation and multiple organ damage in mice [4,5]. Serum biochemical indices are often used to evaluate the physiological state of mice. Transaminases (e.g., alanine aminotransferase (ALT), aspartate aminotransferase (AST)) are a class of enzymes that catalyze the transfer of amino groups between amino acids and keto acids. It is ubiquitously present in animal tissues such as cardiac muscle, brain, liver, and kidney. Under normal conditions, the contents of AST and ALT in serum are low. However, when these tissues (especially liver) develop pathological changes, the corresponding cells are damaged, the permeability of cell membrane increases, and AST and ALT in the cytoplasm are released into the blood [36]. Therefore, the effect of 12% COS on serum aminotransferase levels in mice with LPS-induced endotoxemia were examined. At the same time, the levels of IL-6 and TNF-α in serum were also measured as an evaluation of the degree of inflammatory damage in mice. As shown in Figure 8, the 12% COS could significantly down-regulate the levels of serum biochemical indices and inflammatory cytokines in LPS-induced mice. For ALT, when the concentration is higher than 100 mg/kg, the improvement of ALT is significant and better than the positive control group (dexamethasone sodium phosphate (Dex)). For AST, when the concentration of COS is higher than 50 mg/kg, the improvement effect is already significant. At the same time, the levels of the inflammatory cytokines (IL-6 and TNF-α) in COS groups at various concentrations were lower than the model group. 12% COS can significantly reduce the increase in serum transaminases and pro-inflammatory cytokines caused by LPS and show a concentration-dependent manner. Furthermore, the alleviating effect of 12% COS of 100 mg/kg on AST and IL-6 is better than that of the positive control group (Dex). In addition to these indicators measured in serum, there were also significant changes in the behavior of different groups of mice in the model group and the group with the concentration of 100 mg/kg. The pictures were shown in Appendix A. After the injection of LPS, the mice in the model group developed slow movement, increased eye exudate, and cloudy urine. Mice pretreated with COS at a concentration of 100 mg/kg significantly improved these above phenomena. Therefore, the concentration of 100 mg/kg was selected for histopathological observation experiments to further explore the protective effect of COS on the organs of endotoxemia mice.

After animals were sacrificed, liver and intestine tissues were fixed with 4% paraformaldehyde and observed after H&E staining. Due to the limited information of hepatocytes that H&E slices can provide, the treated liver tissue was also fixed with 2.5% glutaraldehyde and the situation of nuclei and mitochondria in hepatocytes were observed by transmission electron microscopy (TEM). Firstly, the H&E pathological results of liver tissue were shown in Figure 9A. The cells in the NC group were neatly arranged and the cells were plump. However, in the model group (M), the cells were scattered, with many intercellular vacuoles and scattered cytoplasm. These above symptoms were improved in mice pretreated with COS. Through more in-depth observations by TEM, the effect of COS on LPS-induced hepatocyte injury at the subcellular level can be assessed. The nuclear and mitochondrial status of hepatocytes are shown in Figure 9B,C, respectively. The nuclei of the model group had obvious chromatin pyknosis and swelling of the nuclei, and the arrows indicated the large chromatin pyknosis. However, the lesions in the COS group were obviously improved. At the same time, images were used to conduct a semi-quantitative analysis of liver tissue mitochondria according to the Flameng grading method. In the model group (M), the mitochondrial inner cristae disappeared completely, and the mitochondria were swollen and vacant. In contrast, the inner cristae of mitochondria in the COS group could be observed, and the swelling was relieved. The mitochondrial scores of mice in each group are shown in Table 1. As for the intestinal tissue as shown in Figure 10, the intestinal of the model group has shortened villi, deeper crypts, disordered lamina propria in the villi, and unclear cell morphology. Compared with the model group, COS pretreatment increased villi height and the ratio of villus height/crypt depth. The semi-quantitative analysis of the mouse villus state was carried out, and the results are shown in Table 2. Furthermore, the microvilli in the NC group and COS group can be seen under high magnification, while the microvilli in the model group disappear. These results indicate that 12% COS can prevent the organ damage of endotoxemia to a large extent and is expected to become a novel efficient and green anti-inflammatory drug.

## 3. Discussion

As a common clinical disease, endotoxemia has been paid more and more attention. It is caused by excessive invasion of endotoxin, such as LPS. TLR4 recognizes LPS, and binding to LPS leads to cell activation, resulting in the release of pro-inflammatory cytokines. Excessive pro-inflammatory cytokines further lead to system disorders and multiple organ damage. In recent years, the role of COS as an emerging green non-toxic drug in anti-inflammatory has been gradually recognized. For example, the inhibitive effects of COS on LPS-induced IL-6/TNF-α production in macrophages have been explored [37]. Moreover, using differentiated THP-1 cells (human monocyte cells), Paiboon et al. investigated the anti-inflammatory activity of COS and found that COS can reduce the production of multiple pro-inflammatory cytokines associated with LPS [18]. Although many literatures on the anti-inflammatory effect of COS were reported [6,34,38], there has been no systematic explanation of the effect of DA of COS on its anti-inflammatory effect. P. Santos-Moriano et al. once evaluated the differences in the anti-inflammatory effects of COSs of three DAs, fully deacetylated COS, partially acetylated COS and fully acetylated COS, but the DP distribution of these COSs varies widely. The fully deacetylated COS was basically formed by GlcN, (GlcN)_2_, (GlcN)_3_, and (GlcN)_4_, fully acetylated COS contained mostly GlcNAc, (GlcNAc)_2_ and (GlcNAc)_3_, while partially acetylated COS corresponded to a mixture of at least 11 oligosaccharides with different proportions of GlcNAc and GlcN [25]. The inconsistent degree of polymerization makes the DA as a nonunique variable, so the conclusions drawn are not reliable. Therefore, studying the effect of DA on the basis of controlling the DP is necessary for the early development of COS as an efficient and green anti-inflammatory drug.

In this paper, COSs with different DAs were successfully obtained and the DAs were 0%, 12%, 50% and 85%, respectively. The structure of these COSs with different DAs was confirmed by ^1^H NMR and MS analysis, which proved that the DP distribution of our COS samples are consistent, and the DA is the unique variable. These COSs were used to evaluate the effect of DA on the anti-inflammatory activity of COS. By establishing a mouse macrophage inflammation model, the inhibitory effects of COSs with different DAs on LPS-induced inflammation were evaluated from different aspects. We found that DA has an important effect on the anti-inflammatory activity of COS. The anti-inflammatory activity of 12% COS is always better than that of other COSs, including the inhibition of inflammatory cytokines secretion, the downregulation of mRNA of inflammatory cytokine, and the reducing effect on the phosphorylation level of IκBα in NF-κB. Jie Feng et al. studied the receptor-mediated stimulatory effect of COS in macrophages and found that the internalization of COS was mainly attributed to the GlcNAc unit rather than GlcN [39]. Combined with our results, it is speculated that a small amount of acetyl group is also necessary for COS interaction with macrophage to exert a better anti-inflammatory activity. However, the results in this case also showed that the anti-inflammatory activity of COS with high DA exerted decreasing anti-inflammatory activity, even a little pro-inflammatory activity. It is suggested that too much N-acetylation of COS is not favorable to its anti-inflammatory activity.

Furthermore, the results of MS spectra analyzed the detailed oligosaccharides composition. Compared with 0% COS, 12% COS mainly contains some monoacetylated oligosaccharides, such as trisaccharide with one acetyl group (**D2A**), tetrasaccharide with one acetyl group (**D3A**), pentasaccharides with one acetyl group (**D4A**). In contrast, 50% COS and 80% COS mainly contain oligosaccharides with two acetyl groups or even fully N-acetylated oligosaccharides, so it is speculated that these special mono-N-acetylated oligosaccharides present in 12% COS might be responsible for the anti-inflammatory activity of COS. Similar results have been reported in plant priming activity of COS demonstrated by Sven Basa et al. [40]. They proved that mono-acetylated chitotetramers have more remarkably effect in the priming activity of rice cells. At last, we selected 12% COS to evaluate the protective effect of COS on endotoxemia mice. The results showed that the anti-inflammatory activity of COS in vivo was also concentration dependent. Moreover, the COS with the concentration of 100 mg/kg could reduce the elevation of inflammatory cytokines and serum biochemical indices caused by LPS, relieve the behavioral changes caused by LPS in mice and improve the degree of damage to the liver and intestines of mice, which is better than the positive control (Dex).

## 4. Materials and Methods

### 4.1. Reagents

The original COS sample was purchased from Qingdao Yunzhou Biochemistry Co. (Qingdao, China). Fetal bovine serum (FBS) was purchased from Gibco (Carlsbad, CA, USA). RPMI 1640 medium, penicillin and streptomycin were purchased from HyClone (Logan, UT, USA). LPS, 4′,6-diamidino-2-phenylindole (DAPI) and 3-(4,5-dimethyl-2-thiazolyl)-2,5-diphenyl-2-H-tetrazolium bromide (MTT) were purchased from Solarbio Science & Technology Co. (Beijing, China). Antibodies used in WB/immunofluorescence (IF) were purchased from Cell Signaling Technology (Boston, MA, USA). All other chemicals and reagents were of analytical grade. 

### 4.2. Preparation and Characterization of COSs with Different DAs

Several COSs of different DAs were preparate by the N-acetylation of COS. The N-acetylation reaction was modified according to the existing method [41]. Briefly, 1 g of the original COS was dissolved in 250 mL of a methanol/water (50:50, *v*/*v*) solution. Acetic anhydride was added stoichiometrically in the solution under magnetic stirring at room temperature for 1 h. Subsequently, the resulting solution was rotary evaporated to get rid of the methanol and was lyophilized to yield powdered products. By controlling the amount of acetic anhydride added, COSs with different DAs were obtained and analyzed by NMR and MS. To obtain the fully deacetylated COS (DA = 0%) with the same DP distribution, the graded alcohol precipitation method was used on the fully deacetylated COS which is available in our laboratory [42]. Graded alcohol precipitation refers to 10%, 20%, 30%, 40%, 50%, 75% alcohol precipitation of the fully deacetylated COS in sequence, and then centrifuged at 8000 rpm for 15 min, collecting the precipitate individually. HPLC analysis was performed on the precipitates from each step. The DP of the precipitate obtained by 50% ethanol precipitation is 2–6, which is consistent with the DP distribution of other samples (Data is shown in Appendix A) and this sample was selected as COS with DA = 0% in this study. 

The component structures of all COSs were confirmed by ^1^H NMR and MS. The COSs were dissolved in deuterated water and their DAs were analyzed by ^1^H NMR on a JEOL JNM-ECP600 spectrometer. The COSs were analyzed by hydrophilic interaction liquid chromatography using an LC-2030C 3D Plus HPLC system (SHIMADZU, Kyoto, Japan) with an evaporative light scattering detector (Essentia ELSD-16) through the techniques available in our laboratory. The MS detection refers to the method we have published before [43].

### 4.3. Cell Culture and Mouse Model

Macrophages RAW 264.7 cell lines were purchased from the Type Culture Collection of the Chinese Academy of Sciences (Shanghai, China). RAW 264.7 cells were cultured in RPMI 1640 medium supplemented with 10% FBS, 1% double antibiotics (penicillin and streptomycin) in a humidified atmosphere with 5% CO_2_ at 37 °C. In vivo experiments were performed on 4–6-week-old male KM mice weighing 18–22 g purchased from Pengyue Laboratory Animal Breeding Co. (Shandong, China). To maintain environmental quality, all animals were housed in the experimental animal center of Chunghao Tissue Engineering Co. (Shandong, China) at a temperature of 22 ± 1 °C under a 12 h light–dark cycle. All of the animals received care in accordance with the recommendations of the National Institutes of Health Guide for Care and Use of Laboratory Animals and all experimental procedures were approved by the Committee on the Ethics of Animal Experiments of the Institute of Oceanology, Chinese Academy of Sciences.

### 4.4. Experimental Design of Cell and Animal

Cell experiments were grouped as follows: NC group, LPS group, LPS + COS groups with different DAs. The LPS concentration was 1 μg/mL and the COS concentration was 600 μg/mL. (The results of the concentration screening experiment are in Appendix A.) The cells were inoculated into 96-well plates at a density of 1 × 10^5^ cells/mL, cultured for 24 h, and then cultured for another 24 h by changing the medium according to the set groups. After centrifugation at 1200 rpm for 5 min, the supernatant was collected for the determination of NO and inflammatory cytokines. Cells were inoculated into 6-well plates as above and cells in each well were lysed for Real Time Quantitative PCR (RT-qPCR) and WB assays.

The animals were randomly divided into seven groups with 10 mice in each group as follows:

(1) NC group; (2) M: model group; (3) Dexamethasone Sodium Phosphate (Dex): positive control group; (4) 10 mg/kg COS; (5) 50 mg/kg COS; (6) 100 mg/kg COS; (7) 200 mg/kg COS. Mice in COS-treated groups were injected subcutaneously with the corresponding doses of COS three days in advance, and the animals in NC and M groups received an equal volume of vehicle (distilled water) in the same way. Mice were intraperitoneally injected with LPS (5 mg/kg) in the third day to establish an endotoxemia model. The mice were treated with equal-volume saline solution in the NC group. The mice in each group were given free access to food and water. Four hours after the LPS injection, the blood samples were collected by rapid removal of eyeballs before sacrifice and separated by centrifugation at 2000 rpm for 10 min at 4 °C to collect serum samples. Subsequently, all the mice were euthanized, and the liver, intestine and other tissues were harvested and immediately stored at −80 °C [44].

### 4.5. Determination of Cell Viability

The viability of RAW264.7 cells was measured by MTT assay as described previously [45]. Briefly, cells were seeded into a 96-well plate, the cell density is 1 × 10^5^ cells/mL and cultured for 24 h, and then exposed to different concentrations of COSs. Afterward, RPMI 1640 medium containing MTT (0.5 mg/mL) was added to each well and incubated for another 4 h at 37 °C. After removing the MTT solution, the formazan was dissolved with 150 μL DMSO. The absorbance at 490 nm was read on a microplate reader. The experiment set 6 replicates for each sample.

### 4.6. Biochemical Index Determination and Enzyme-Linked Immunosorbent Assay

For the determination of NO content, cell culture supernatant was mixed 1:1 with prepared Griess reagent. The OD_540 nm_ was measured with a microplate reader, and the NO content was calculated according to the NaNO_2_ standard curve, with 6 replicate wells for each sample. Inflammatory cytokine levels in the cell supernatant and mice serum were quantified using ELISA kits (Beijing Solarbio Science & Technology Co., Beijing, China) specified for mouse TNF-α, IL-6 according to the manufacturer’s instructions. ALT, AST levels in the serum were quantified using corresponding assay kits from Rayto Life and Analytical Sciences Co. These indicators were measured in three replicates and data are presented as means ± SD of three replicates.

### 4.7. RT-qPCR and WB Analysis

Total RNA was extracted from sample-treated cells using RNA-Solv Reagent. Changes in the steady-state concentration of mRNA for IL-6, TNF-α, IL-1β, iNOS, COX-2 and NOX2 were assessed by RT-qPCR. The extracted RNA was quantified with Nanodrop 2000 for subsequent inversion. Evo M-MLV Reversal Kit was used to remove gDNA, and then reverse transcription experiment was performed. Using this cDNA as a template, take 2 μL of this cDNA stock solution and use SYBR Green Pro Taq HS premixed qPCR kit for qPCR amplification to detect gene expression level. The primer sequences for Real-time qPCR were shown in Table 3.

The culture medium was discarded, and the cells were washed with PBS, and then the cell lysis solution containing protease inhibitors and phosphatase inhibitor was added; the cells were fully lysed on ice for 30 min, and the concentration of protein was determined by BCA kit for quantification. The proteins were analyzed on 8–12% SDS-PAGE. The proteins were blotted on an immunoblot PVDF membrane and blocked with 5% skim milk in TBST containing 0.1% Tween 20. Then, the corresponding primary antibody and the secondary antibody were incubated. After the incubation was completed, blots were developed with the ECL detection kit and analyzed grayscale by image J software.

### 4.8. Observing p65 Nuclear Translocation by IF

Nuclear translocation of NF-κB (p65) was analyzed by immunofluorescence. The cells were cultured in a 6-well plate pre-installed with sterile cell slides, and the cells were treated according to the experimental design. The medium was discarded and fixed with 4% paraformaldehyde for 15 min at room temperature. PBS washed three times. Blocked with 1 × PBS containing 5% goat serum, 0.3% Triton X-100 for 1 h at room temperature. The primary antibody against p65 and secondary antibody rabbit anti-goat IgG/FITC were diluted with 1 × PBS containing 1% BSA, 0.3% Triton X-100. The cells were incubated with primary antibody overnight at 4 °C, followed by incubation with rabbit anti-goat IgG/FITC antibody. Then, the cells were washed by PBS three times. Nuclei were stained with DAPI. After washing with PBS, the slides were mounted with anti-fluorescence quenching mounting medium, and the nuclear entry of p65 was observed under a confocal laser microscope.

### 4.9. Histological Analysis

After animals were sacrificed, liver and intestine tissues were fixed with 4% paraformaldehyde, embedded in paraffin, stained with hematoxylin solution for 3–5 min, rinsed with deionized water. After dehydration with graded ethanol, the cells were stained with eosin dye. After dehydration, the samples were observed with a microscope, and the images were collected and analyzed. The treated liver tissue was fixed with 2.5% glutaraldehyde and post-fixed with 1% osmium tetroxide. After being dehydrated and embedded, Representative areas were chosen for ultrathin sectioning and observed by TEM [46]. According to the Flameng grading method [47], semi-quantitative analysis of liver tissue mitochondria was performed using TEM images. Fields of view were randomly selected from each group. Mitochondria were counted and the degree of damage was scored (0–4) in each field. Finally, the score of all mitochondria in the fields was divided by the number of mitochondria. The semi-quantitative scoring standard was as follows. 0 point: mitochondrial morphology and structure were completely normal; 1 point: the mitochondrial structure was normal, part of the mitochondria was slightly swollen and the mitochondrial ridge was separated; 2 points: the mitochondria were severely swollen and the ridge of mitochondria was separated but not broken; 3 points: mitochondria were severely swollen and a large number of mitochondrial ridges were broken; 4 points: mitochondria were vacuolated, extremely swollen, the ridge was broken or even disappeared and the inner and outer membrane was broken.

Villus height and crypt depth of stained sections were measured microscopically using a Nikon ECLIPSE 80i optical microscope equipped with a computer-aided morphometric system (Nikon Corporation, Tokyo, Japan). For each intestinal sample, a total of 10–15 well-oriented and intact villi were measured.

### 4.10. Statistical Analysis

The data are presented as the mean ± SD, followed by Duncan’s multiple-range tests. Differences were considered to be statistically significant if *p* < 0.05.

## 5. Conclusions

In this study, we prepared four COSs with different DAs (0%, 12%, 50% and 85%, respectively) and the same DP. The structure of these COSs was confirmed by ^1^H NMR and MS analysis, which proved that the DP distribution of these COSs is consistent, and the DA is the unique variable. The anti-inflammatory activity of these COSs with different DAs were studied and the DA was found to be an important parameter influencing the anti-inflammatory activity of COS. The anti-inflammatory activity of 12% COS is always better than that of other COSs and we selected 12% COS to evaluate the protective effect on endotoxemia mice. The results show that 12% COS can effectively prevent the inflammatory response of endotoxemia in mice. In conclusion, the influence of the important structural parameter (DA) of COS has been deeply studied on its anti-inflammatory activity, which will be beneficial to the further utilization of this marine oligosaccharide. At the same time, the mouse endotoxemia model again confirmed the good anti-inflammatory effect of COS in vivo, indicating that COS is expected to become a safer and more efficient anti-inflammatory drug in the near future.

## Figures and Tables

**Figure 1 ijms-23-08205-f001:**
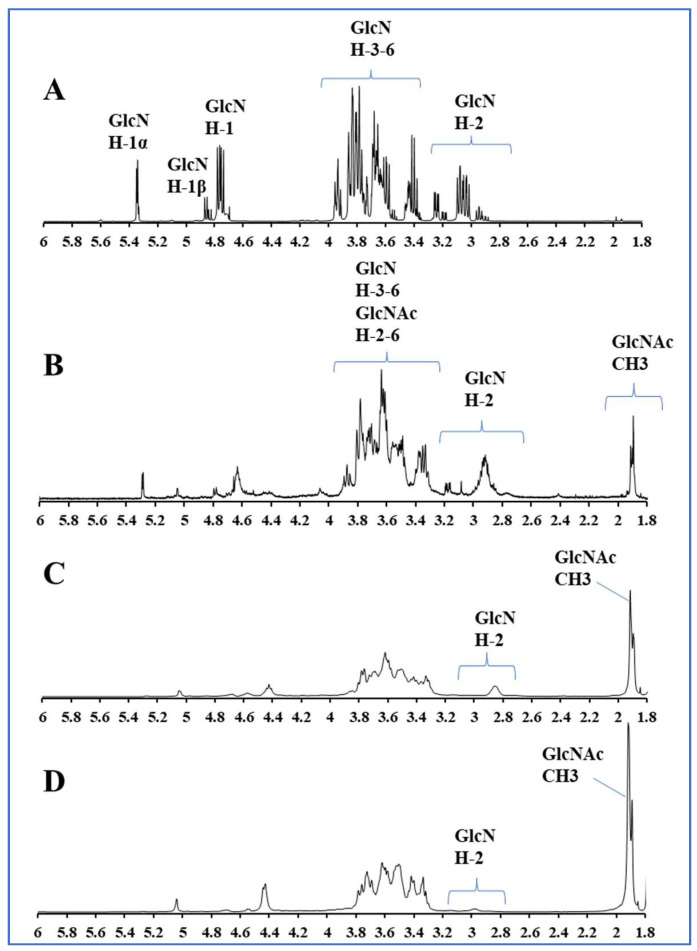
^1^H NMR spectra of COSs with different DAs. (**A**–**D**) represent 0% COS, 12% COS, 50% COS and 85% COS, respectively. The peak at 2.8–3.2 ppm is the hydrogen on the second carbon of GlcN, and the peak at 1.9 ppm is the hydrogen on the acetyl group in GlcNAc. The peak intensity at 1.9 ppm increases linearly with DA increasing.

**Figure 2 ijms-23-08205-f002:**
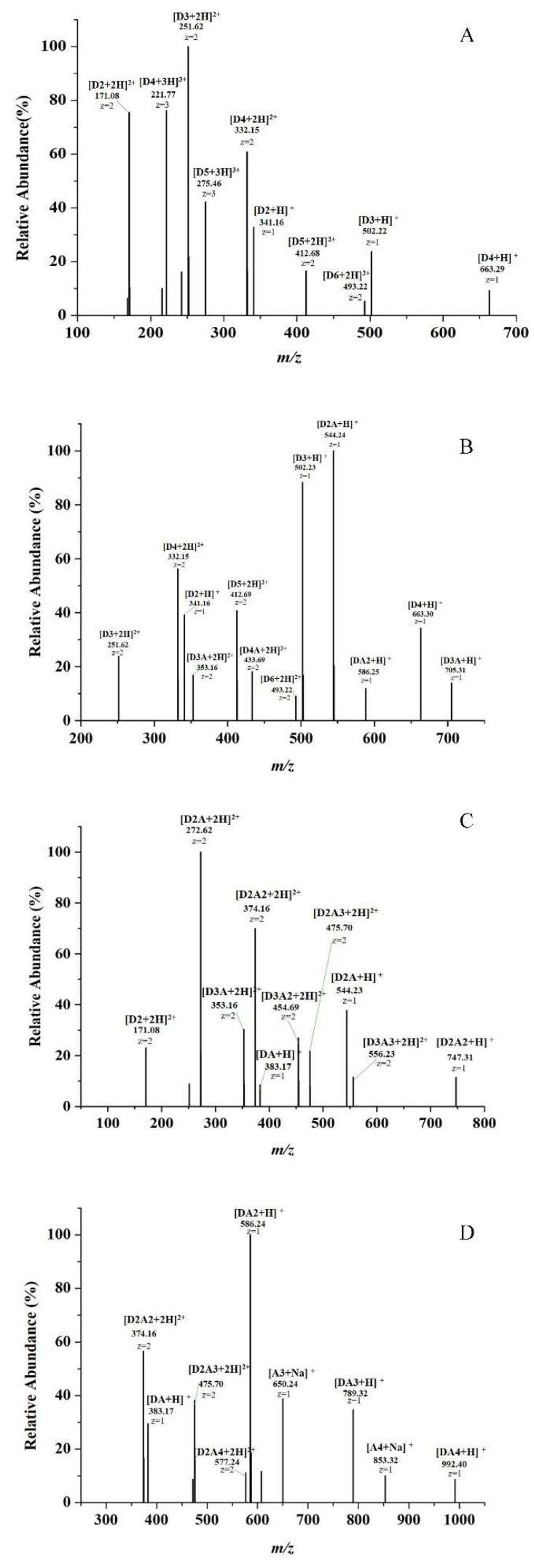
Mass spectrum analysis of the four prepared COSs with different DAs. (**A**) The positive-ion mode ESI-MS spectrum of COS with DA = 0%, (**B**) The positive-ion mode ESI-MS spectrum of COS with DA = 12%, (**C**) The positive-ion mode ESI-MS spectrum of COS with DA = 50%, (**D**) The positive-ion mode ESI-MS spectrum of COS with DA = 85%. The numbers in the figure only represent the number of monosaccharide units.

**Figure 3 ijms-23-08205-f003:**
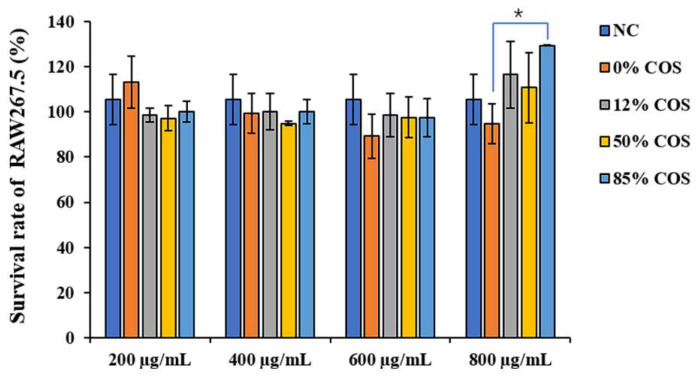
Effects of different concentrations of COSs on macrophage cell viability. At the concentrations used in this study, these COSs had no cytotoxic effects on cells. The data are presented as means ± SD of three replicates. * There is a significant difference between 0% COS with 85% COS in 800 μg/mL, *p* ≤ 0.05.

**Figure 4 ijms-23-08205-f004:**
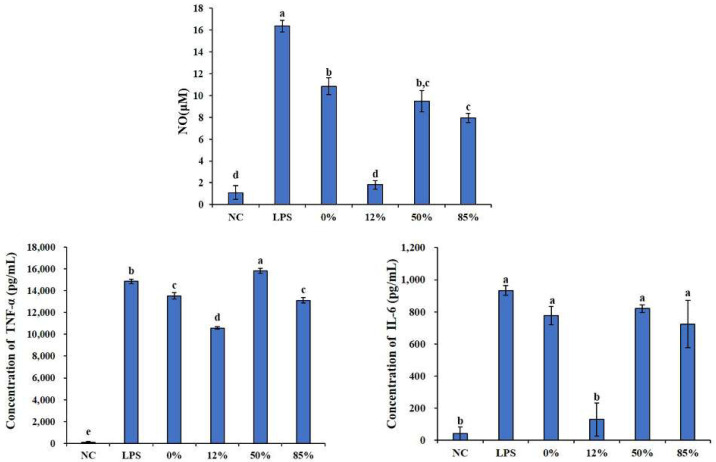
COSs with different DAs inhibited LPS-induced secretion of NO, IL-6 and TNF-α. 0%, 12%, 50%, 85% represent COS with DA of 0%, 12%, 50%, 85%, respectively. The data are presented as means ± SD of three replicates. From “a” to “e”, the mean values decrease in turn. There is no significant difference between groups with the same letter. Different letters indicate individual groups for multiple comparisons with significant differences (one-way ANOVA, Duncan, *p* < 0.05).

**Figure 5 ijms-23-08205-f005:**
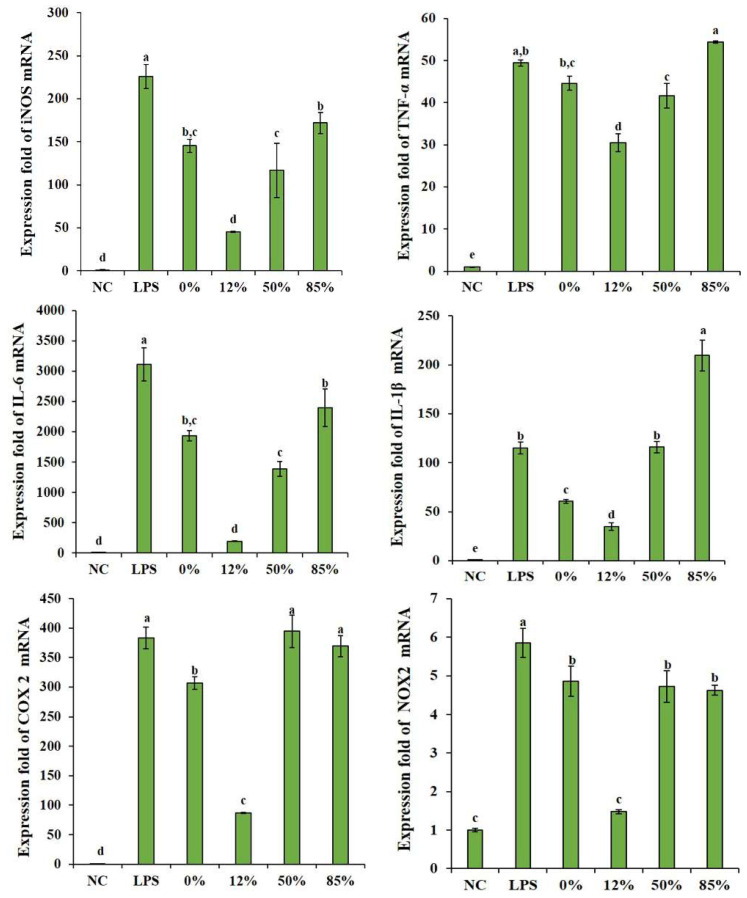
COS can reduce the up-regulation of several inflammation-related genes caused by LPS. 0%, 12%, 50%, 85% represent COS with DA of 0%, 12%, 50%, 85%, respectively. The data are presented as means ± SD of three replicates. From “a” to “e”, the mean values decrease in turn. There is no significant difference between groups with the same letter. Different letters indicate individual groups for multiple comparisons with significant differences (one-way ANOVA, Duncan, *p* < 0.05).

**Figure 6 ijms-23-08205-f006:**
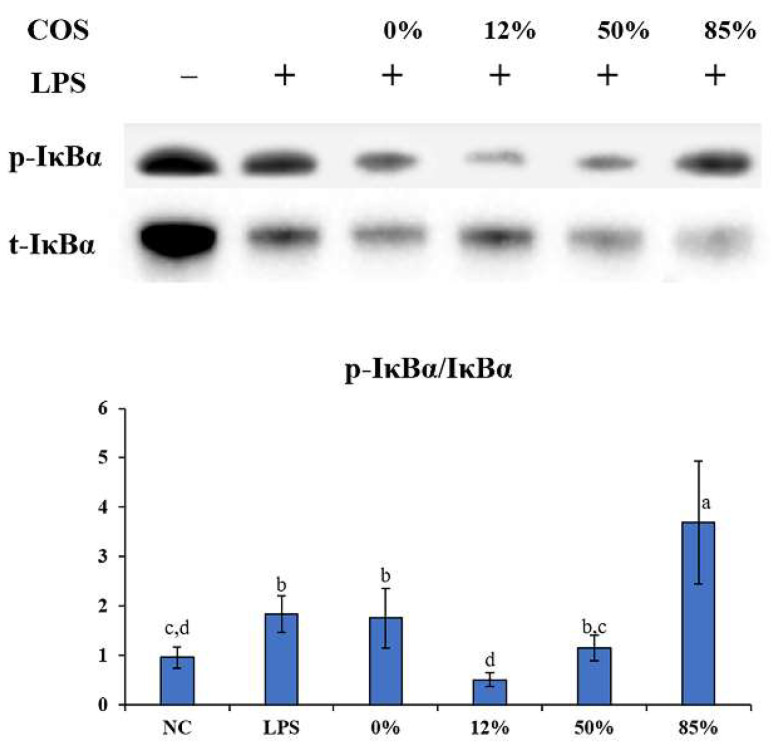
Western blotting of phosphorylated-IκBα and IκBα. After the cells were lysed, the total protein was extracted for WB experiments. 0%, 12%, 50%, 85% represent COS with DA of 0%, 12%, 50%, 85%, respectively. The data are presented as means ± SD of three replicates. From “a” to “d”, the mean values decrease in turn. There is no significant difference between groups with the same letter. Different letters indicate individual groups for multiple comparisons with significant differences (one-way ANOVA, Duncan, *p* < 0.05).

**Figure 7 ijms-23-08205-f007:**
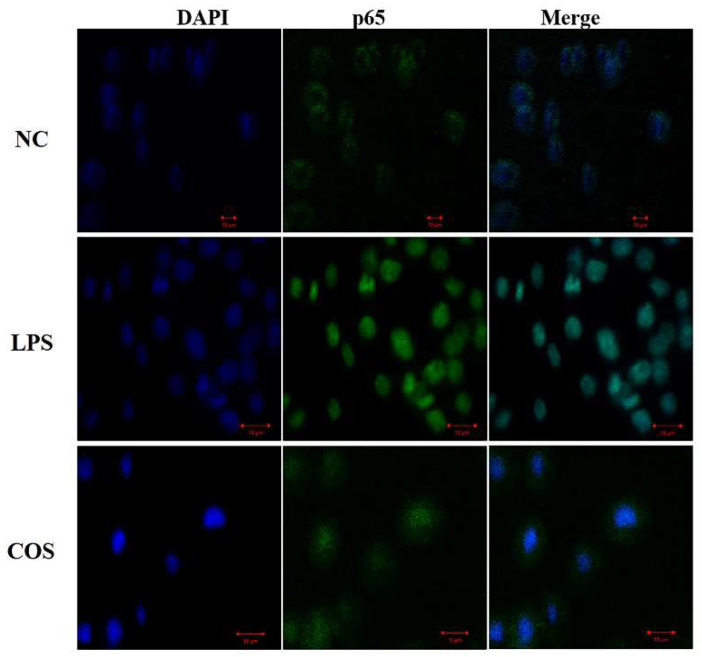
The immunofluorescent staining assay was used to analyze the nuclear translocation of NF-κB in RAW264.7 cells. NF-κB was stained with rabbit anti-NF-κB p65 IgG, with FITC-conjugated goat anti-rabbit IgG as secondary antibody (green). Nuclei were stained with DAPI (blue). Measuring bar 10 μm.

**Figure 8 ijms-23-08205-f008:**
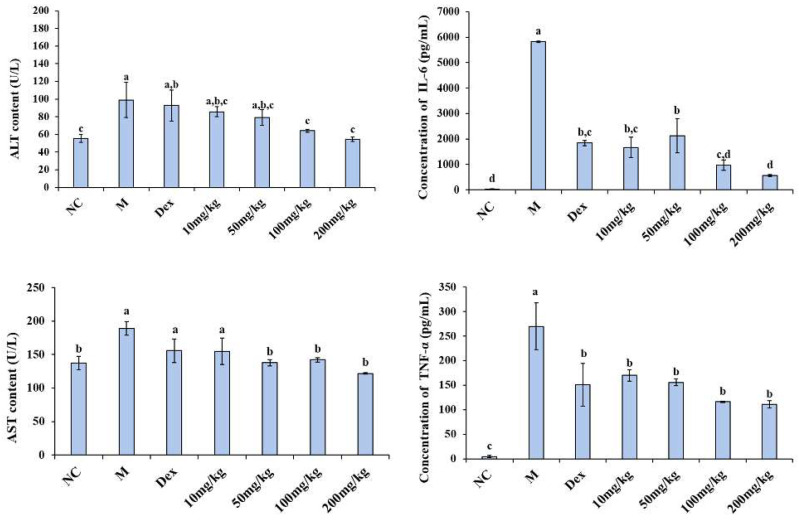
Changes in pro-inflammatory markers and transaminases with 12% COS treatment. M represents the model group; Dex represents the positive control group. The data are presented as means ± SD of three replicates. From “a” to “e”, the mean values decrease in turn. There is no significant difference between groups with the same letter. Different letters indicate individual groups for multiple comparisons with significant differences (one-way ANOVA, Duncan, *p* < 0.05).

**Figure 9 ijms-23-08205-f009:**
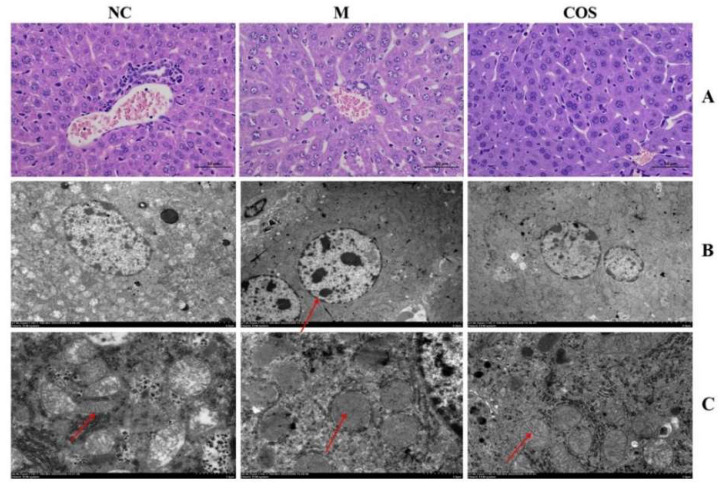
H&E staining and TEM of liver tissue. (**A**) represents the H&E staining of liver tissue; (**B**) represents the nuclei of hepatocytes under the same magnification. The arrow indicates chromatin condensation; (**C**) represents the hepatocyte mitochondria under the same magnification. Arrows indicate mitochondrial inner cristae.

**Figure 10 ijms-23-08205-f010:**
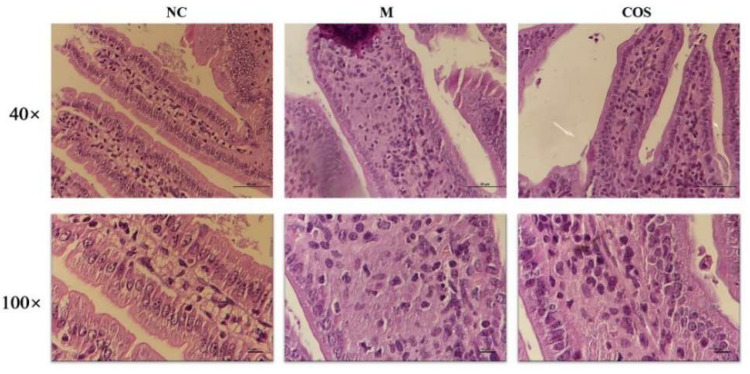
H&E staining of small intestinal villi. Compared with the model group, COS pretreatment increased villi height and the microvilli in the NC group and COS group can be seen under high magnification, while the microvilli in the model group disappear.

**Table 1 ijms-23-08205-t001:** The mitochondrial scores of mice.

Group	NC	M	COS
Score	1.0125 ± 0.15519 ^a^	3.5625 ± 0.14752 ^c^	1.4571 ± 0.11722 ^b^

^a,b,c^: The data are presented as means ± SD (*n* = 8–10). Different alphabets were used to indicate significant differences between groups if *p* < 0.05.

**Table 2 ijms-23-08205-t002:** The ratio of villus height/crypt depth.

Group	NC	M	COS
Height/crypt depth	3.49 ± 0.10179 ^a^	1.85 ± 0.089 ^c^	2.21 ± 0.090 ^b^

^a,b,c^: The data are presented as means ± SD (*n* = 10–15). Different alphabets were used to indicate significant differences between groups if *p* < 0.05.

**Table 3 ijms-23-08205-t003:** The primer sequences for Real-time qPCR.

Primer.	Forward Primer Sequence (5′–3′)	Reverse Primer Sequence (5′–3′)
GAPDH	ACTCACGGCAAATTCAACGGCA	GACTCCACGACATACTCAGCAC
iNOS	GCCCAGGAGGAGAGAGAT	GCAAAGAGGACTGCGGCT
IL-6	AGACTTCCATCCAGTTGCCTTCTTG	CATGTGTAATTAAGCCTCCGACTTGTG
TNF-α	TGCCTATGTCTCAGCCTCTTC	GAGGCCATTTGGGAACTTCT
IL-1β	GCAGAGCACAAGCCTGTCTTCC	ACCTGTCTTGGCCGAGGACTAAG
COX-2	AAATGCTGGTGTGGAAGGT	TTGTTGCTCTAGGCTTTGCT
NOX2	CCTTACTGGCTGGGATGA	GCAATGGTCTTGAACTCGT

## Data Availability

Not applicable.

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
