# Peer review of "The Effect of N-Acetylation on the Anti-Inflammatory Activity of Chitooligosaccharides and Its Potential for Relieving Endotoxemia"

_ijms, 2022, doi:10.3390/ijms23158205_

Round 1

Reviewer 1 Report

The authors of the manuscript entitled: "The effect of N-acetylation on the anti-inflammatory activity of chitooligosaccharides and its potential for relieving endotoxemia" prepared several acetylated chitooligosaccharides. Their degree of acetylation was verified by NMR. Subsequently, they investigated their biological activity in an LPS-induced inflammation model. Overall, the manuscript is readable, well structured with minimal typographical errors.

I have the following comments and observations on the manuscript:

1) Read the text carefully and edit/consolidate abbreviations, introductions and use of already established abbreviations. The abstract is separate and abbreviations are not part of the text itself. For example: line 60 you used COS at the first time without explaining what it is; line 67 used DA at the first time; line 46 you used TNF-a, but in line 78-79 you introduce this abbreviation; For a wider readership, I would also introduce abbreviations for GlcN and others.

2) The figure descriptions need to be improved. The current legends are insufficient. Figure legends should contain information so that the reader knows what they are and what they mean, e.g. signification, and does not have to look through the text to find out what is what.

3) The results in Figure 6 are from only one repetition? If not, you need to add the deviations and the number of repetitions. If yes, then you need to either measure the repetitions to calculate significance or state in the results that you are speculating based on one repetition... Because you can't conclude from one repetition, only speculate.

4) Separate lines 271 and 272 with another line. Currently, figure legend 8 merges with the MS text itself.

5) I would recommend to put the Conclusion section right after the 3rd discussion section and then put the Methods material. Currently the Material Methods section breaks up the readability of the manuscript.

6) In the methodology, unify the writing of RAW 264.7. You combine RAW and Raw, now.

7) Sections 4.8, 4.9 and 4.10 need to be block-aligned like other text.

8) I would add more sentences to the Conclusion section where this finding could be used, potential future impact.

Overall, I rate this manuscript as a success and after answering and editing the text, according to my comments, it will be suitable for publication. The topic of oligo- and polysaccharides is interesting and brings many potential applications. As an example, I recommend publications dealing with isolated natural polysaccharides that have a modulating effect on immune cells and, in combination with their probiotic properties, are suitable as an addition to the treatment of intestinal and immune diseases.

Georgiev et al. (2022) Structural characterization of polysaccharides from Geranium sanguineum L. and their immunomodulatory effects in response to inflammatory agents.

Georgiev et al. (2017) Acidic polysaccharide complexes from purslane, silver linden and lavender stimulate Peyer's patch immune cells through innate and adaptive mechanisms

Georgiev et al. (2017) Tilia tomentosa pectins exhibit dual mode of action on phagocytes as β-glucuronic acid monomers are abundant in their rhamnogalacturonans I

Georgiev et al. (2017) The common lavender (Lavandula angustifolia Mill.) pectic polysaccharides modulate phagocytic leukocytes and intestinal Peyer's patch cells

Author Response

1)    Read the text carefully and edit/consolidate abbreviations, introductions and use of already established abbreviations. The abstract is separate and abbreviations are not part of the text itself. For example: line 60 you used COS at the first time without explaining what it is; line 67 used DA at the first time; line 46 you used TNF-a, but in line 78-79 you introduce this abbreviation; For a wider readership, I would also introduce abbreviations for GlcN and others.

Reply: Thank you very much for taking your time to read our article and give us the comments. We've double checked/ consolidate abbreviations and made changes in line 46 (IL, TNF), line 60 (COS), line 67(DA), line 101 (GlcN) and line 103,139,187 etc. Thanks again for your suggestions.

2)    The figure descriptions need to be improved. The current legends are insufficient. Figure legends should contain information so that the reader knows what they are and what they mean, e.g. signification, and does not have to look through the text to find out what is what.

Reply: Thanks for giving us the comments. We added more figure descriptions in figure 3,4,6,8 and 10 (in line 166-169, 205-209, 255-259, 298-300, 342-343). For figure 3,4,5,6,8, we added the description of     statistical differences which each alphabet indicates. For figure 3,4,10, we've made it clearer what these figures mean.

3)    The results in Figure 6 are from only one repetition? If not, you need to add the deviations and the number of repetitions. If yes, then you need to either measure the repetitions to calculate significance or state in the results that you are speculating based on one repetition... Because you can't conclude from one repetition, only speculate.

Reply: Thanks for your question. The results in Figure 6 are not from one repetition. The figure shows the average results, we have added the standard error and statistical analysis in line 254 to replace the previous figure.

4)    Separate lines 271 and 272 with another line. Currently, figure legend 8 merges with the MS text itself

Reply: Thanks for your advice. We have added another line in that place as shown in line 300-301.

5)    I would recommend to put the Conclusion section right after the 3rd discussion section and then put the Methods material. Currently the Material Methods section breaks up the readability of the manuscript.

Reply: Thanks for your advice. We have adjusted the layout of these two parts according to your suggestion. Thanks again for your suggestions.

6)    In the methodology, unify the writing of RAW 264.7. You combine RAW and Raw, now.

Reply: Thanks for your advice. We are very sorry for this error and we have now corrected this error in line 460.

7)    Sections 4.8, 4.9 and 4.10 need to be block-aligned like other text

Reply: Thanks for your advice. We are very sorry for this error and we have now corrected this error in line sections 5.8, 5.9 and 5.10.(since we put the Conclusion section right after the 3rd discussion section and then put the Methods material as the fifth part according to your suggestion.)

8)    I would add more sentences to the Conclusion section where this finding could be used, potential future impact

Reply: Thanks for your advice. We added more sentences to the Conclusion section of where this finding could be used and the great potential of COS as an anti-inflammatory drug in line 415-420.

In addition, we have benefited a lot from reading the articles you provided, and cited them in this article in line 63-64. Thanks again for your review.

Reviewer 2 Report

Story is reasonable. However, there are several concerns to confirm the result of present study as follows. 

1. Average and standard deviation of negative control (data derived from cells which were cultured without COS) should be shown in Fig3. In addition, statistical analysis should be checked. Cells treated with 800 μg/mL of 85% COS shows obviously higher survival rate. 

2. Statistical differences which each alphabet indicates should be clearly described in figure legend of Fig 4, 5 and 8. 

3. There are no error bar in Fig 6. Average and standard deviation of this experiments should be shown to prove the universality and reproducibility of the experiment. Statistical analysis is also necessary. 

4. Is the data showed in Fig 6 derived from in vitro experiment? The origin of data should be clearly described in the text and figure legend. 

5. The result posted on Fig 7 should be quantified. Counting number of cells which were translocated into nuclei seems to be most simple analysis. Statistical analysis of quantified data is also need. Similarly, quantification and statistical analysis are necessary to the data showing as Fig 9 and 10. Authorized damage scoring seems to be available. 

6. Negative control or other description would be better instead of CK. 

7. Resolution of each figure is low. Some characters and words are not easy to read.

Author Response

1)  Average and standard deviation of negative control (data derived from cells which were cultured without COS) should be shown in Fig3. In addition, statistical analysis should be checked. Cells treated with 800 μg/mL of 85% COS shows obviously higher survival rate.

Reply: Thanks for giving us the comments. We have added the  average and standard deviation of negative control in figure 3 to replace the previous figure in line 165.

As for statistical analysis, we have done a significant analysis before. There is no significant difference between each group and the negative control group (in line 154).

The results indicated that these COSs had no cytotoxic effect on cells at the concentrations used in this study. At a high concentration of 800μg/mL, there is a significant difference between the 0% group and 85% group. But it is not much of significance for the cytotoxic results of COSs.

2)  Statistical differences which each alphabet indicates should be clearly described in figure legend of Fig 4, 5 and 8.

Reply: Thanks for giving us the comments. We added more figure descriptions in figure 3,4,5,6,8 (in line 166-169, 205-209, 215-217, 255-259, 298-300). For these figures we added the description of   statistical differences which each alphabet indicates. By the way, we've added more descriptions for figure 3,4,10.

3)  There are no error bar in Fig 6. Average and standard deviation of this experiments should be shown to prove the universality and reproducibility of the exper-iment. Statistical analysis is also necessary.

Reply: Thanks for your question. The results in Figure 6 are not from one repetition. The figure shows the average results, we have added the standard error and statistical analysis in line 254 to replace the previous figure.

4)  Is the data showed in Fig 6 derived from in vitro experiment? The origin of data should be clearly described in the text and figure legend.

Reply: Thanks for your question and comments. The data showed in Fig 6 is surely derived from in vitro experiment, we added some details in line 229-230 and line 255-259. Thanks again for your suggestions.

5)  The result posted on Fig 7 should be quantified. Counting number of cells which were translocated into nuclei seems to be most simple analysis. Statistical analysis of quantified data is also need. Similarly, quantification and statistical analysis are necessary to the data showing as Fig 9 and 10. Authorized damage scoring seems to be available.

Reply: Thanks for giving us the comments. For Figure 7, we didn't count the cells at the experiments time, but our aim was to do a qualitative analysis, mainly looking at the localization of p65 protein. It may be our negligence. Although the number of cells is not counted, our results are sufficient to prove that 12%COS could effectively inhibiting p65 entry into the nucleus to a certain extent. We appreciate your suggestion and will improve it when it comes to such experiments later. And for Figure 9.10, we did do some work before and now we have added it into the modified version, including experimental methods in line 551-566 and experimental results in line 305-307,309-320,324-325. As there are already many figures in this study, we present our results in tabular form as shown in line 336-338, 345-347.

6)  Negative control or other description would be better instead of CK.

Reply: Thanks for your advice. We have used negative control (NC) to replace CK according to your suggestion and commented it on line 154. At the same time, we checked through the full text and revised all the figures, tables and texts. Thank you very much for your suggestions.

7)  Resolution of each figure is low. Some characters and words are not easy to read.

Reply: Thanks for your advice. We have further checked the figures in the full text. We have enlarged the annotations and characters in all figures and change some layout, such as figure 4.5.8. In addition, we will provide all the original figures to the magazine with a resolution higher than 300dpi.

Round 2

Reviewer 1 Report

The authors answered all my questions and improved the suggestions I didn't make. Therefore, I am satisfied with the current state of the manuscript and recommend it for publication.

Author Response

Reviewer #1:

The authors answered all my questions and improved the suggestions I didn't make. Therefore, I am satisfied with the current state of the manuscript and recommend it for publication.

Reply: Thanks for taking your time to review our article and providing useful comments.

Reviewer 2 Report

Three points still should be considered as follows.

1. In Fig. 3, any marking is necessary to indicate statistical difference between the 0% group and 85% group since the difference is statistically significant. In addition, this is not recognized as slight differences since 20% or more enhancement in cell growth is observed in 85% COS (800μg/mL) -treated group compared to NC group. Similar tendency is observed in the case of 12% COS and 50% COS. Hence, the reason should be described in the text.

2. What is the “n=3” means in Fig.3,4,5 and 6? Repetition of experiment? Triplicated? This point should be described method section or anywhere clearly. 

3. Did the previous comment about alphabet used for indication of statistical differences make sense? Which does alphabet indicates which groups of statistical differences? For instance, what does alphabet "a" indicate? There is no explanation about the meaning of each alphabet so that whether COS treatment is effective or not cannot be evaluated as a reviewer. Please make all statistical differences between groups clear in each figure as like the way that every reader can understand.

Author Response

  1.  In Fig. 3, any marking is necessary to indicate statistical difference between the 0% group and 85% group since the difference is statistically significant. In addition, this is not recognized as slight differences since 20% or more enhancement in cell growth is observed in 85% COS (800μg/mL) -treated group compared to NC group. Similar tendency is observed in the case of 12% COS and 50% COS. Hence, the reason should be described in the text.

Reply: Thank you very much for taking your time to read our article and give us the comments. As your said, marking is necessary to indicate statistical difference between the 0%COS group and 85%COS group since the difference is statistically significant. Correspondingly, we have added the * mark indicating statistical difference and added more description for the legend of this figure in line 169-172. In addition, as the enhancement in cell growth you pointed out really exists, high concentrations of chitosan or chitooligosaccharides could cause cell proliferation, and there are many literatures reporting on this phenomenon [Zhai et al., 2018, Xu et al., 2021]. However, there is no significant difference between these groups and the NC group except for the group of 85% COS at 800μg/mL. We took this situation into account in our experiment and chose the concentration of 600μg/mL in this study to avoid the effects of proliferation, and described the reason in line 157, 163-166. Overall, the concentration of COSs used in this study had neither cytotoxic effects or promoting proliferation on cells. Thanks again for your suggestions.

1).          Zhai, X. C.; Yang, X.; Zou, P.; Shao, Y.; Yuan, S. J.; Abd El-Aty, A. M.; Wang, J., Protective Effect of Chitosan Oligosaccharides Against Cyclophosphamide-Induced Immunosuppression and Irradiation Injury in Mice. J. Food Sci. 2018, 83, (2), 535-542.

2).          Xu, C.; Xing, R.; Liu, S.; Qin, Y.; Li, K.; Yu, H.; Li, P., Immunostimulatory effect of N-2-hydroxypropyltrimethyl ammonium chloride chitosan-sulfate chitosan complex nanoparticles on dendritic cells. Carbohydr Polym 2021, 251, 117098.

  1. What is the “n=3” means in Fig.3,4,5 and 6? Repetition of experiment? Triplicated? This point should be described method section or anywhere clearly. 

Reply: Thanks for giving us the comments. n=3 means that we measured three parallel samples, which meet the biological requirements. We explain this in detail in the Methods section, as lines 514-515, and we make changes to the way the legends are described as shown in line 169-172, 208-211, 215-219, 257-262, 301-304.  

  1.  Did the previous comment about alphabet used for indication of statistical differences make sense? Which does alphabet indicates which groups of statistical differences? For instance, what does alphabet "a" indicate? There is no explanation about the meaning of each alphabet so that whether COS treatment is effective or not cannot be evaluated as a reviewer. Please make all statistical differences between groups clear in each figure as like the way that every reader can understand.

Reply: Thanks for giving us the comments. Since there are many groups used in this article, and at the same time, we need to compare each other pairwise. Traditional notations, such as “*” or “**”, seem hard to make the figures very clear. In this study, we used the one-way ANOVA, Duncan test to show the statistical differences with different alphabets. This method has been very popular in many journals in recent years. [Almaguer-Melian et al., 2012; Jing et al., 2022; Ye et al., 2022], which is more suitable to label the statistical differences of multiple goups.

The steps are roughly as follows:

  • Arrange all the means in descending order.
  • Mark the letter “a” on the largest mean, and compare the mean marked with “a” with the following means; if the difference is not significant, mark it as “a”; when a certain mean that is significantly different from it occurs, mark it as “b”.
  • Then take the mean marked with “b” as the standard, and compare it with the means that are larger than it. Any value that is not significant will be marked as “b”, and then take the maximum mean marked with “b” as the standard. Compared this mean with the following unmarked means, those that are not significant will continue to be marked as “b”, until there is a mean that is significantly different from it, marked as “c”.
  • Repeat this until the smallest mean has a marker letter and is compared with all the above means.
  • In this way, there is no significant difference between groups with the same letter. Different letters indicate individual groups for multiple comparisons with significant differences (p<0.05).

We explain in more detail in the legends of figure 4.5.6.8, such as line 208-211, 215-219, 257-262, 301-304.  

1).          Jing, H.; Korasick, D. A.; Emenecker, R. J.; Morffy, N.; Wilkinson, E. G.; Powers, S. K.; Strader, L. C., Regulation of AUXIN RESPONSE FACTOR condensation and nucleo-cytoplasmic partitioning. Nature Communications 2022, - 13, (- 1).

2).         Almaguer-Melian, W.; Bergado-Rosado, J.; Pavón-Fuentes, N.; Alberti-Amador, E.; Mercerón-Martínez, D.; Frey, J. U., Novelty exposure overcomes foot shock-induced spatial-memory impairment by processes of synaptic-tagging in rats. PNAS 2012, 109, (3), 953-958.

3).          Ye, N.; Han, W.; Toseland, A.; Wang, Y.; Fan, X.; Xu, D.; van Oosterhout, C.; Aslam, S. N.; Barry, K.; Beszteri, B.; Brussaard, C.; Clum, A.; Copeland, A.; Daum, C.; Duncan, A.; Eloe-Fadrosh, E.; Fong, A.; Foster, B.; Foster, B.; Ginzburg, M.; Huntemann, M.; Ivanova, N. N.; Kyrpides, N. C.; Martin, K.; Moulton, V.; Mukherjee, S.; Palaniappan, K.; Reddy, T. B. K.; Roux, S.; Schmidt, K.; Strauss, J.; Timmermans, K.; Tringe, S. G.; Underwood, G. J. C.; Valentin, K. U.; van de Poll, W. H.; Varghese, N.; Grigoriev, I. V.; Tagliabue, A.; Zhang, J.; Zhang, Y.; Ma, J.; Qiu, H.; Li, Y.; Zhang, X.; Mock, T.; Sea of Change, C., - The role of zinc in the adaptive evolution of polar phytoplankton. Nature Ecology & Evolution 2022, - 6, (- 7), - 978.

Round 3

Reviewer 2 Report

The manuscript was well revised.